# PERCEPTUAL GENERATIVE AUTOENCODERS

**Zijun Zhang**
University of Calgary

**Ruixiang Zhang**
MILA, Université de Montréal

**Zongpeng Li**
Wuhan University

**Yoshua Bengio**
MILA, Université de Montréal
CIFAR Senior Fellow

**Liam Paull**
MILA, Université de Montréal
CIFAR AI Chair

## ABSTRACT

Modern generative models are usually designed to match target distributions directly in the data space, where the intrinsic dimensionality of data can be much lower than the ambient dimensionality. We argue that this discrepancy may contribute to the difficulties in training generative models. We therefore propose to map both the generated and target distributions to the latent space using the encoder of a standard autoencoder, and train the generator (or decoder) to match the target distribution in the latent space. The resulting method, perceptual generative autoencoder (PGA), is then incorporated with maximum likelihood or variational autoencoder (VAE) objective to train the generative model. With maximum likelihood, PGA generalizes the idea of reversible generative models to unrestricted neural network architectures and arbitrary latent dimensionalities. When combined with VAE, PGA can generate sharper samples than vanilla VAE.

## 1 INTRODUCTION

Recent years have witnessed great interest in generative models, mainly due to the success of generative adversarial networks (GANs) (Goodfellow et al., 2014; Radford et al., 2016; Karras et al., 2018; Brock et al., 2019). Despite the prevalence, the adversarial nature of GANs can lead to a number of challenges, such as unstable training dynamics and mode collapse. Since the advent of GANs, substantial efforts have been devoted to addressing these challenges (Salimans et al., 2016; Arjovsky et al., 2017; Gulrajani et al., 2017; Miyato et al., 2018), while non-adversarial approaches that are free of these issues have also gained attention. Examples include variational autoencoders (VAEs) (Kingma & Welling, 2014), reversible generative models (Dinh et al., 2017; Kingma & Dhariwal, 2018), and Wasserstein autoencoders (WAEs) (Tolstikhin et al., 2018).

However, non-adversarial approaches often have significant limitations. For instance, VAEs tend to generate blurry samples, while reversible generative models require restricted neural network architectures or solving neural differential equations (Grathwohl et al., 2019). Furthermore, to use the change of variable formula, the latent space of a reversible model must have the same dimensionality as the data space, which is unreasonable considering that real-world, high-dimensional data (e.g., images) tends to lie on low-dimensional manifolds, and thus results in redundant latent dimensions and variability. Intriguingly, recent research (Arjovsky et al., 2017; Dai & Wipf, 2019) suggests that the discrepancy between the intrinsic and ambient dimensionalities of data also contributes to the difficulties in training GANs and VAEs.

In this work, we present a novel framework for training autoencoder-based generative models, with non-adversarial losses and unrestricted neural network architectures. Given a standard autoencoder and a target data distribution, instead of matching the target distribution in the data space, we map both the generated and target distributions to the latent space using the encoder, and train the generator (or decoder) to minimize the divergence between the mapped distributions. We prove, under mild assumptions, that by minimizing a form of latent reconstruction error, matching the target distribution in the latent space implies matching it in the data space. We call this framework *perceptual generative autoencoder (PGA)*. We show that PGA enables training generative autoencoders with

maximum likelihood, without restrictions on architectures or latent dimensionalities. In addition, when combined with VAE, PGA can generate sharper samples than vanilla VAE.[1]

## 2 METHODS

### 2.1 PERCEPTUAL GENERATIVE MODEL

Let $f : \mathbb{R}^D \to \mathbb{R}^H$ be the encoder parameterized by $\phi$, and $g : \mathbb{R}^H \to \mathbb{R}^D$ be the decoder parameterized by $\theta$. Our goal is to obtain a generative model, which maps a simple prior distribution to the data distribution, $\mathcal{D}$. Throughout this paper, we use $\mathcal{N}(\mathbf{0}, \mathbf{I})$ as the prior distribution.

For $\mathbf{z} \in \mathbb{R}^H$, the output of the decoder, $g(\mathbf{z})$, lies in a manifold that is at most $H$-dimensional. Therefore, if we train the autoencoder to minimize

$$L_r = \frac{1}{2} \mathbb{E}_{\mathbf{x} \sim \mathcal{D}} \left[ \|\hat{\mathbf{x}} - \mathbf{x}\|_2^2 \right], \tag{1}$$

where $\hat{\mathbf{x}} = g(f(\mathbf{x}))$, then $\hat{\mathbf{x}}$ can be seen as a projection of the input data, $\mathbf{x}$, onto the manifold of $g(\mathbf{z})$. Let $\hat{\mathcal{D}}$ denote the distribution of $\hat{\mathbf{x}}$. Given enough capacity of the encoder, $\hat{\mathcal{D}}$ is the best approximation to $\mathcal{D}$ (in terms of L2 distance), that we can obtain from the decoder, and thus can serve as a surrogate target for training the generator.

Due to the difficulty of directly training the generator to match $\hat{\mathcal{D}}$, we seek to map $\hat{\mathcal{D}}$ to the latent space, and train the generator to match the mapped distribution, $\hat{\mathcal{H}}$, in the latent space. To this end, we reuse the encoder for mapping $\hat{\mathcal{D}}$ to $\hat{\mathcal{H}}$, and train the generator such that $h(\cdot) = f(g(\cdot))$ maps $\mathcal{N}(\mathbf{0}, \mathbf{I})$ to $\hat{\mathcal{H}}$. In addition, to ensure that $g$ maps $\mathcal{N}(\mathbf{0}, \mathbf{I})$ to $\hat{\mathcal{D}}$, we minimize the following latent reconstruction loss with respect to (w.r.t.) $\phi$:

$$L_{lr,\mathcal{N}}^{\phi} = \frac{1}{2} \mathbb{E}_{\mathbf{z} \sim \mathcal{N}(\mathbf{0}, \mathbf{I})} \left[ \|h(\mathbf{z}) - \mathbf{z}\|_2^2 \right]. \tag{2}$$

Formally, let $Z(\mathbf{x})$ be the set of all $\mathbf{z}$'s that are mapped to the same $\mathbf{x}$ by the decoder, we have the following theorem:

**Theorem 1.** *Assuming the convexity of $Z(\mathbf{x})$ for all $\mathbf{x} \in \mathbb{R}^D$, and sufficient capacity of the encoder; for $\mathbf{z} \sim \mathcal{N}(\mathbf{0}, \mathbf{I})$, if Eq. (2) is minimized and $h(\mathbf{z}) \sim \hat{\mathcal{H}}$, then $g(\mathbf{z}) \sim \hat{\mathcal{D}}$.*

*Proof.* We first show that any different $\mathbf{x}$'s generated by the decoder are mapped to different $\mathbf{z}$'s by the encoder. Let $\mathbf{x}_1 = g(\mathbf{z}_1)$, $\mathbf{x}_2 = g(\mathbf{z}_2)$, and $\mathbf{x}_1 \neq \mathbf{x}_2$. Since the encoder has sufficient capacity and Eq. (2) is minimized, we have $f(\mathbf{x}_1) = \mathbb{E}[\mathbf{z}_1]$ and $f(\mathbf{x}_2) = \mathbb{E}[\mathbf{z}_2]$. By definition, $\mathbf{z}_1 \in Z(\mathbf{x}_1)$, $\mathbf{z}_2 \in Z(\mathbf{x}_2)$. Therefore, given the convexity of $Z(\mathbf{x}_1)$ and $Z(\mathbf{x}_2)$, $f(\mathbf{x}_1) \in Z(\mathbf{x}_1)$ and $f(\mathbf{x}_2) \in Z(\mathbf{x}_2)$. Since $Z(\mathbf{x}_1) \cap Z(\mathbf{x}_2) = \varnothing$, we have $f(\mathbf{x}_1) \neq f(\mathbf{x}_2)$.

For $\mathbf{z} \sim \mathcal{N}(\mathbf{0}, \mathbf{I})$, denote the distributions of $g(\mathbf{z})$ and $h(\mathbf{z})$, respectively, by $\widetilde{\mathcal{D}}$ and $\widetilde{\mathcal{H}}$. We then consider the case where $\widetilde{\mathcal{D}}$ and $\hat{\mathcal{D}}$ are discrete distributions. If $g(\mathbf{z}) \nsim \hat{\mathcal{D}}$, then there exists an $\mathbf{x}$, such that $p_{\widetilde{\mathcal{H}}}(f(\mathbf{x})) = p_{\widetilde{\mathcal{D}}}(\mathbf{x}) \neq p_{\hat{\mathcal{D}}}(\mathbf{x}) = p_{\hat{\mathcal{H}}}(f(\mathbf{x}))$, contradicting that $h(\mathbf{z}) \sim \hat{\mathcal{H}}$. The result still holds when $\widetilde{\mathcal{D}}$ and $\hat{\mathcal{D}}$ approach continuous distributions. □

Note that the two distributions compared in Theorem 1, $\widetilde{\mathcal{D}}$ and $\hat{\mathcal{D}}$, are mapped respectively from $\mathcal{N}(\mathbf{0}, \mathbf{I})$ and $\mathcal{H}$. While $\mathcal{N}(\mathbf{0}, \mathbf{I})$ is supported on the whole $\mathbb{R}^H$, there can be $\mathbf{z}$'s with low probabilities in $\mathcal{N}(\mathbf{0}, \mathbf{I})$, but with high probabilities in $\mathcal{H}$, which are not well covered by Eq. (2). Therefore, it is sometimes helpful to minimize another latent reconstruction loss on $\mathcal{H}$:

$$L_{lr,\mathcal{H}}^{\phi} = \frac{1}{2} \mathbb{E}_{\mathbf{z} \sim \mathcal{H}} \left[ \|h(\mathbf{z}) - \mathbf{z}\|_2^2 \right]. \tag{3}$$

By Theorem 1, the problem of training the generative model reduces to training $h$ to map $\mathcal{N}(\mathbf{0}, \mathbf{I})$ to $\hat{\mathcal{H}}$, which we refer to as the perceptual generative model. In the subsequent subsections, we present a maximum likelihood approach, as well as a VAE-based approach, to train the perceptual generative model.

---

[1]Code is available at https://github.com/zj10/PGA.

## 2.2 A Maximum Likelihood Approach

We first assume the invertibility of $h$. For $\hat{\mathbf{x}} \sim \hat{\mathcal{D}}$, let $\hat{\mathcal{H}}$ be the distribution of $f(\hat{\mathbf{x}})$. We can train $h$ directly with maximum likelihood using the change of variable formula as

$$\mathbb{E}_{\hat{\mathbf{z}} \sim \hat{\mathcal{H}}} \left[ \log p(\hat{\mathbf{z}}) \right] = \mathbb{E}_{\mathbf{z} \sim \mathcal{H}} \left[ \log p(\mathbf{z}) - \log \left| \det \left( \frac{\partial h(\mathbf{z})}{\partial \mathbf{z}} \right) \right| \right]. \tag{4}$$

Ideally, we would like to maximize Eq. (4) w.r.t. the parameters of the generator (or decoder), $\theta$. However, directly optimizing the first term in Eq. (4) requires computing $\mathbf{z} = h^{-1}(\hat{\mathbf{z}})$, which is usually unknown. Nevertheless, for $\hat{\mathbf{z}} \sim \hat{\mathcal{H}}$, we have $h^{-1}(\hat{\mathbf{z}}) = f(\mathbf{x})$ and $\mathbf{x} \sim \mathcal{D}$, and thus we can minimize the following loss function w.r.t. $\phi$ instead:

$$L_{nll}^{\phi} = -\mathbb{E}_{\mathbf{z} \sim \mathcal{H}} \left[ \log p(\mathbf{z}) \right] = \frac{1}{2} \mathbb{E}_{\mathbf{x} \sim \mathcal{D}} \left[ \| f(\mathbf{x}) \|_2^2 \right]. \tag{5}$$

To avoid computing the Jacobian for the second term in Eq. (4), which is slow for unrestricted architectures, we approximate the Jacobian determinant and derive a loss function for the decoder as

$$L_{nll}^{\theta} = \frac{H}{2} \mathbb{E}_{\mathbf{z} \sim \mathcal{H}, \delta \sim \mathcal{S}(\epsilon)} \left[ \log \frac{\| h(\mathbf{z} + \delta) - h(\mathbf{z}) \|_2^2}{\| \delta \|_2^2} \right] \approx \mathbb{E}_{\mathbf{z} \sim \mathcal{H}} \left[ \log \left| \det \left( \frac{\partial h(\mathbf{z})}{\partial \mathbf{z}} \right) \right| \right], \tag{6}$$

where $\mathcal{S}(\epsilon)$ is a uniform distribution on a small hypersphere of radius $\epsilon$. When $\epsilon \to 0$, the approximation forms an upper bound on the right-hand side (r.h.s.) of Eq. (6), and becomes tight if $h$ is close to the identity function. Intuitively, Eq. (5) attracts the latent representations of data samples to the origin, while Eq. (6) expands the volume occupied by each sample in the latent space.

The above discussion relies on the assumption that $h$ is invertible, which is not necessarily true for unrestricted architectures. If $h(\mathbf{z})$ is not invertible for some $\mathbf{z}$, the logarithm of the Jacobian determinant at $\mathbf{z}$ becomes infinite, in which case Eq. (4) cannot be optimized. Nevertheless, since $\| h(\mathbf{z} + \delta) - h(\mathbf{z}) \|_2^2$ is unlikely to be zero if the model is properly initialized, the approximation in Eq. (6) remains finite, and thus can be optimized regardless.

To summarize, we train the autoencoder to obtain a generative model by minimizing the following loss function:

$$L = L_r + \alpha L_{lr,\mathcal{N}}^{\phi} + \beta L_{lr,\mathcal{H}}^{\phi} + \gamma \left( L_{nll}^{\phi} + L_{nll}^{\theta} \right), \tag{7}$$

where $\alpha$, $\beta$, and $\gamma$ are hyperparameters to be tuned. We refer to this approach as maximum likelihood PGA (LPGA).

## 2.3 A VAE-based Approach

The original VAE is trained by maximizing the evidence lower bound on $\log p(\mathbf{x})$ as

$$\begin{aligned} \log p(\mathbf{x}) &\geq \log p(\mathbf{x}) - \mathbb{KL}(q(\mathbf{z} \mid \mathbf{x}) \parallel p(\mathbf{z} \mid \mathbf{x})) \\ &= \mathbb{E}_{\mathbf{z} \sim q(\mathbf{z} \mid \mathbf{x})} \left[ \log p(\mathbf{x} \mid \mathbf{z}) \right] - \mathbb{KL}(q(\mathbf{z} \mid \mathbf{x}) \parallel p(\mathbf{z})), \end{aligned} \tag{8}$$

where $p(\mathbf{x} \mid \mathbf{z})$ is modeled with the decoder, and $q(\mathbf{z} \mid \mathbf{x})$ is modeled with the encoder. In our case, we would like to modify Eq. (8) in a way that helps maximize $\log p(\hat{\mathbf{z}})$. Therefore, we replace $p(\mathbf{x} \mid \mathbf{z})$ on the r.h.s. of Eq. (8) with $p(\hat{\mathbf{z}} \mid \mathbf{z})$, and derive a lower bound on $\log p(\hat{\mathbf{z}})$ as

$$\begin{aligned} \log p(\hat{\mathbf{z}}) &\geq \log p(\hat{\mathbf{z}}) - \mathbb{KL}(q(\mathbf{z} \mid \mathbf{x}) \parallel p(\mathbf{z} \mid \hat{\mathbf{z}})) \\ &= \mathbb{E}_{\mathbf{z} \sim q(\mathbf{z} \mid \mathbf{x})} \left[ \log p(\hat{\mathbf{z}} \mid \mathbf{z}) \right] - \mathbb{KL}(q(\mathbf{z} \mid \mathbf{x}) \parallel p(\mathbf{z})). \end{aligned} \tag{9}$$

Similar to the original VAE, we make the assumption that $q(\mathbf{z} \mid \mathbf{x})$ and $p(\hat{\mathbf{z}} \mid \mathbf{z})$ are Gaussian; i.e., $q(\mathbf{z} \mid \mathbf{x}) = \mathcal{N} \left( \mathbf{z} \mid \mu_\phi(\mathbf{x}), \mathrm{diag} \left( \sigma_\phi^2(\mathbf{x}) \right) \right)$, and $p(\hat{\mathbf{z}} \mid \mathbf{z}) = \mathcal{N} \left( \hat{\mathbf{z}} \mid \mu_{\theta,\phi}(\mathbf{z}), \sigma^2 \mathbf{I} \right)$. Here, $\mu_\phi(\cdot) = f(\cdot)$, $\mu_{\theta,\phi}(\cdot) = h(\cdot)$, and $\sigma > 0$ is a tunable scalar. The VAE variant is trained by minimizing

$$L_{vae} = -\mathbb{E}_{\mathbf{x} \sim \mathcal{D}} \left[ \mathbb{E}_{\mathbf{z} \sim q(\mathbf{z} \mid \mathbf{x})} \left[ \log p(\hat{\mathbf{z}} \mid \mathbf{z}) \right] - \mathbb{KL}(q(\mathbf{z} \mid \mathbf{x}) \parallel p(\mathbf{z})) \right]. \tag{10}$$

Note that we slightly abuse the notation, since $\mathbf{z}$ is deterministic for the losses in Eqs. (2) and (3), but is stochastic (with additive Gaussian noise) in Eq. (10). Accordingly, the overall loss function is given by

$$L = L_r + \alpha L_{lr,\mathcal{N}}^{\phi} + \beta L_{lr,\mathcal{H}}^{\phi} + \gamma L_{vae}. \tag{11}$$

We refer to this approach as variational PGA (VPGA).

## 3 EXPERIMENTS

In this section, we evaluate the performance of LPGA and VPGA on three image datasets, MNIST (LeCun et al., 1998), CIFAR-10 (Krizhevsky & Hinton, 2009), and CelebA (Liu et al., 2015). For CelebA, we employ the discriminator and generator architecture of DCGAN (Radford et al., 2016) for the encoder and decoder of PGA. We half the number of filters (i.e., $64$ filters for the first convolutional layer) for faster experiments, while more filters are observed to improve performance. Due to smaller input sizes, we reduce the number of convolutional layers accordingly for MNIST and CIFAR-10, and add a fully-connected layer of $1024$ units for MNIST, as done in (Chen et al., 2016). SGD with a momentum of $0.9$ is used to train all models.

The training process of PGA is stable in general, given the non-adversarial losses. However, stability issues can occur when batch normalization (Ioffe & Szegedy, 2015) is introduced, since both the encoder and decoder are fed with multiple batches drawn from different distributions. In our experiments, we only use batch normalization when it does not cause stability issues, in which case it is observed to substantially accelerate convergence.

As shown in Fig. 1, the visual quality of the PGA-generated samples is significantly improved over that of VAE. In particular, VPGA generates much sharper samples on CIFAR-10 and CelebA compared to vanilla VAE. For CelebA, we further show latent space interpolations in Fig. 2. In addition, we use the Fréchet Inception Distance (FID) (Heusel et al., 2017) to evaluate LPGA, VPGA, and VAE. For each model and each dataset, we take 5,000 generated samples to compute the FID score. The results are summarized in Table. 1. Compared to other non-adversarial generative models (Tolstikhin et al., 2018; Kolouri et al., 2019; Ghosh et al., 2019), where similar but larger architectures are used, we obtain substantially better FID scores on CIFAR-10 and CelebA.

Table 1: FID scores of LPGA, VPGA, and VAE.

| Model | MNIST | CIFAR-10 | CelebA |
|-------|-------|----------|--------|
| LPGA | $12.06 \pm 0.12$ | $55.87 \pm 0.25$ | $\mathbf{14.53} \pm 0.52$ |
| VPGA | $\mathbf{11.67} \pm 0.21$ | $\mathbf{51.51} \pm 1.16$ | $24.73 \pm 1.25$ |
| VAE | $15.55 \pm 0.18$ | $115.74 \pm 0.63$ | $43.60 \pm 0.33$ |

## 4 CONCLUSION

We proposed a framework, PGA, for training autoencoder-based generative models, with non-adversarial losses and unrestricted neural network architectures. By matching target distributions in the latent space, PGA trained with maximum likelihood generalizes the idea of reversible generative models to unrestricted neural network architectures and arbitrary latent dimensionalities. In addition, it improves the performance of VAE when combined together.

In principle, PGA can be combined with any method that can train the perceptual generative model. While we have only considered two non-adversarial approaches, an interesting future work would be to combine PGA with an adversarial discriminator trained on latent representations. Moreover, the compatibility issue with batch normalization deserves further investigation.

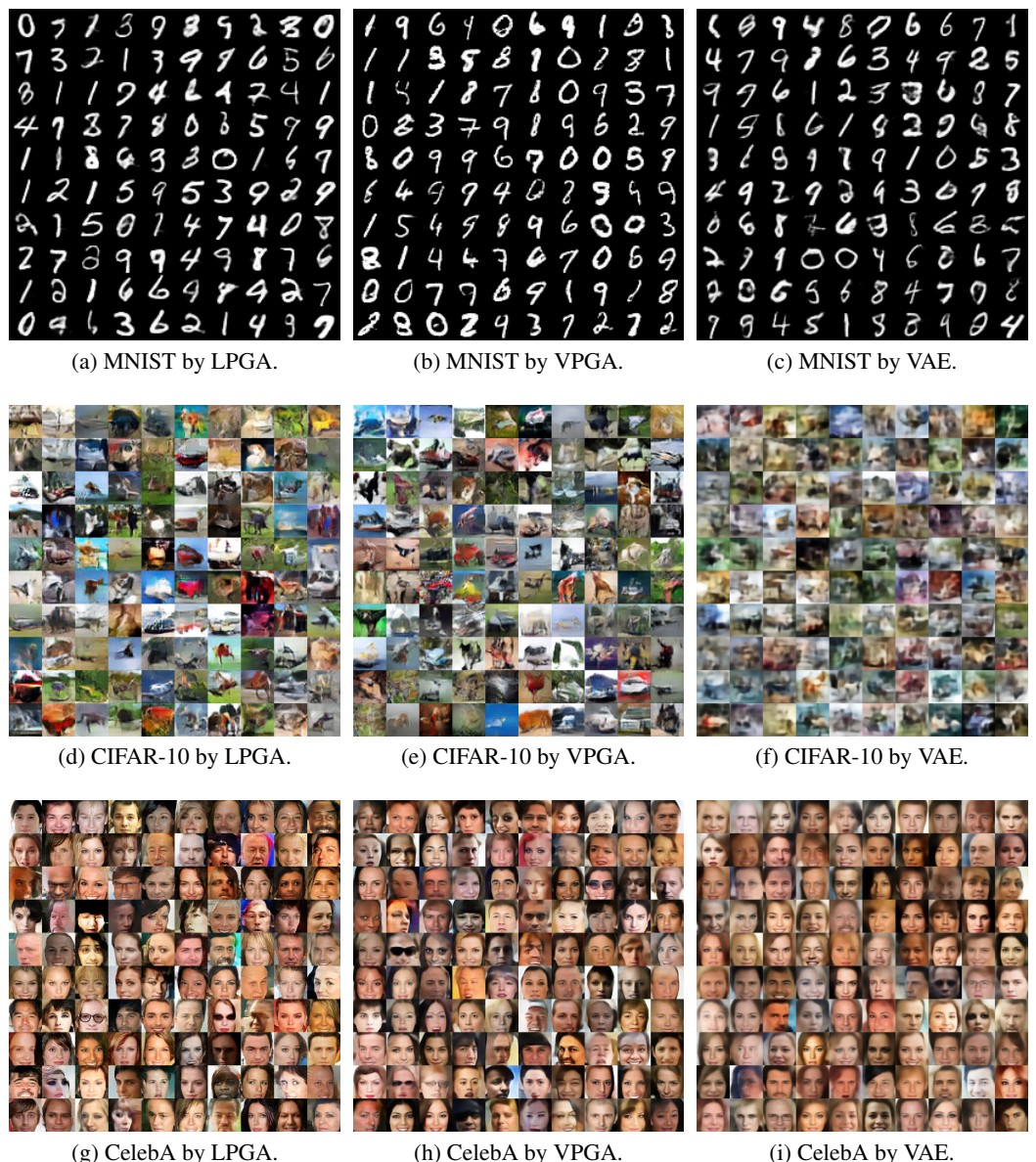

(a) MNIST by LPGA.  (b) MNIST by VPGA.  (c) MNIST by VAE.

(d) CIFAR-10 by LPGA.  (e) CIFAR-10 by VPGA.  (f) CIFAR-10 by VAE.

(g) CelebA by LPGA.  (h) CelebA by VPGA.  (i) CelebA by VAE.

Figure 1: Random samples generated by LPGA, VPGA, and VAE.

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

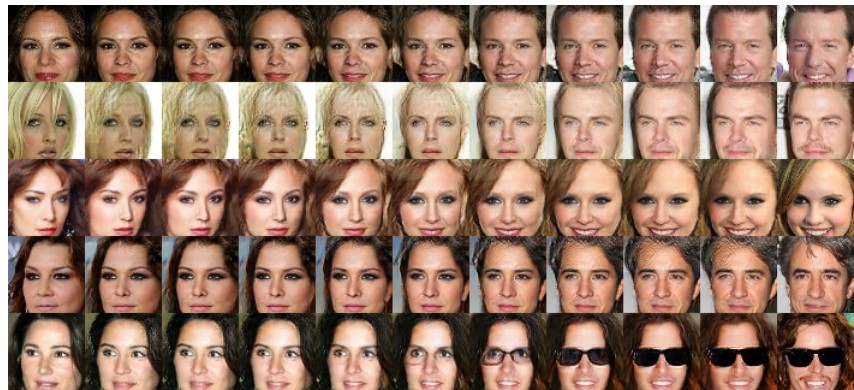

(a) Interpolations generated by LPGA.

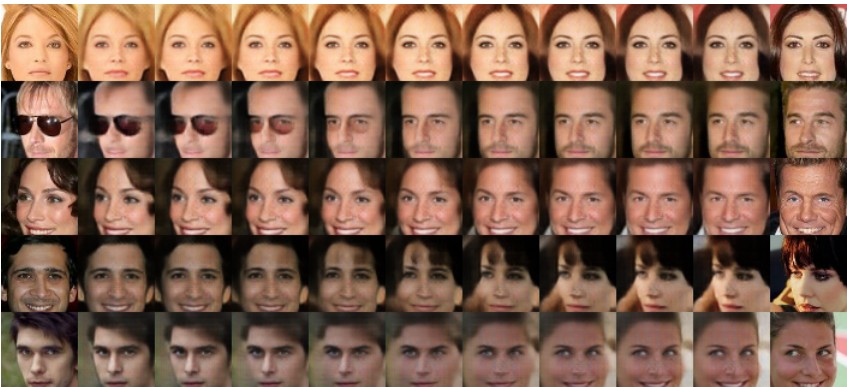

(b) Interpolations generated by VPGA.

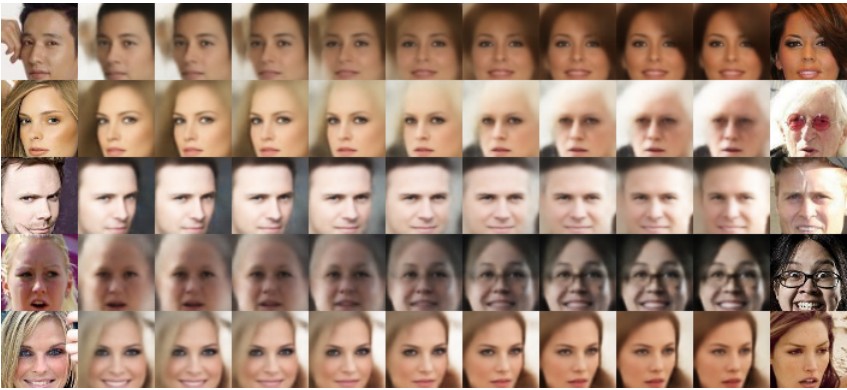

(c) Interpolations generated by VAE.

Figure 2: Latent space interpolations on CelebA.

Laurent Dinh, Jascha Sohl-Dickstein, and Samy Bengio. Density estimation using real nvp. In *International Conference on Learning Representations*, 2017.

Partha Ghosh, Mehdi S. M. Sajjadi, Antonio Vergari, Michael Black, and Bernhard Schölkopf. From variational to deterministic autoencoders. *arXiv preprint arXiv:1903.12436*, 2019.

Ian Goodfellow, Jean Pouget-Abadie, Mehdi Mirza, Bing Xu, David Warde-Farley, Sherjil Ozair, Aaron Courville, and Yoshua Bengio. Generative adversarial nets. In *Advances in neural information processing systems*, pp. 2672–2680, 2014.

Will Grathwohl, Ricky TQ Chen, Jesse Betterncourt, Ilya Sutskever, and David Duvenaud. Ffjord: Free-form continuous dynamics for scalable reversible generative models. In *International Conference on Learning Representations*, 2019.

Ishaan Gulrajani, Faruk Ahmed, Martin Arjovsky, Vincent Dumoulin, and Aaron C Courville. Improved training of wasserstein gans. In *Advances in Neural Information Processing Systems*, pp. 5767–5777, 2017.

Martin Heusel, Hubert Ramsauer, Thomas Unterthiner, Bernhard Nessler, and Sepp Hochreiter. Gans trained by a two time-scale update rule converge to a local nash equilibrium. In *Advances in Neural Information Processing Systems*, pp. 6626–6637, 2017.

Sergey Ioffe and Christian Szegedy. Batch normalization: Accelerating deep network training by reducing internal covariate shift. In *International Conference on Machine Learning*, pp. 448–456, 2015.

Tero Karras, Timo Aila, Samuli Laine, and Jaakko Lehtinen. Progressive growing of gans for improved quality, stability, and variation. In *International Conference on Learning Representations*, 2018.

Diederik P Kingma and Max Welling. Auto-encoding variational bayes. In *International Conference on Learning Representations*, 2014.

Durk P Kingma and Prafulla Dhariwal. Glow: Generative flow with invertible 1x1 convolutions. In *Advances in Neural Information Processing Systems*, pp. 10236–10245, 2018.

Soheil Kolouri, Phillip E Pope, Charles E Martin, and Gustavo K Rohde. Sliced wasserstein autoencoders. In *International Conference on Learning Representations*, 2019.

Alex Krizhevsky and Geoffrey Hinton. Learning multiple layers of features from tiny images. Technical report, University of Toronto, 2009.

Yann LeCun, Corinna Cortes, and Chris J. C. Burges. The mnist handwritten digit database, 1998.

Ziwei Liu, Ping Luo, Xiaogang Wang, and Xiaoou Tang. Deep learning face attributes in the wild. In *International Conference on Computer Vision*, 2015.

Takeru Miyato, Toshiki Kataoka, Masanori Koyama, and Yuichi Yoshida. Spectral normalization for generative adversarial networks. In *International Conference on Learning Representations*, 2018.

Alec Radford, Luke Metz, and Soumith Chintala. Unsupervised representation learning with deep convolutional generative adversarial networks. In *International Conference on Learning Representations*, 2016.

Tim Salimans, Ian Goodfellow, Wojciech Zaremba, Vicki Cheung, Alec Radford, and Xi Chen. Improved techniques for training gans. In *Advances in neural information processing systems*, pp. 2234–2242, 2016.

Ilya Tolstikhin, Olivier Bousquet, Sylvain Gelly, and Bernhard Schoelkopf. Wasserstein autoencoders. In *International Conference on Learning Representations*, 2018.

