# OpenReview forum: "Perceptual Generative Autoencoders"
_ICLR.cc/2019/Workshop/DeepGenStruct — DeepGenStruct 2019_

### Official Review · AnonReviewer1 · 2019-04-05
**Convoluted, but an interesting idea**

**Rating:** 3
**Confidence:** 3

**Review:**

This paper relies on autoencoders in order to to distribution matching in high dimensional spaces, with the following recipe:
   1) build an autoencoder of the data g(f(x)) - eq 1.
   2)  build an autoencoder in latent space - ensure that f(g(z)) can reconstruct both z samples from N(0, 1) - eq 2 and f(g(f(x))) - eq 3.
   3) Show (under assumptions) that if  eq 2 is minimized, for  z samples from N(0, 1), if f(g(z)) = f(g(f(x)) for some x in the original data space, then g(z) belongs to the reconstruction distribution. This entails that if the conditions of the theorem are satisfied (namely if f(g(z)) = f(g(f(x)) for some x in the original data space). sample quality will match reconstruction quality.
  4) Use 3) to justify that  f(g(z)) should have high likelihood in the distribution induced by f(g(x)). Achieve that either by maximum likelihood (approximate - since there is no guarantee that h is invertible) or by variational inference.

Equation 5: equation 5 follows from a change of variable and then using that f(x) for x in the data will be normally distributed. What ensures this? Minimizing equation (2) ensures that f(g(z)) will be normally distribution, with z sampled from N(0, 1).

Cons:
  * The paper is very convoluted to read. Notation is missing and the discussion is missing important aspects which is making following the correctness of the paper difficult. I urge the authors to add further discussions and figures.
  * Parts of the loss function used are rather ad hoc.
 * The method seems dependent on 3 important hyperparameters. The sensitivity to hyperparameters is not discussed.

Pros:
  * good empirical results
  * code is open sourced

Citations: I would add a citation to VEEGAN[1] which also uses distribution matching in latent space.

[1] Srivastava, Akash, et al. "Veegan: Reducing mode collapse in gans using implicit variational learning." Advances in Neural Information Processing Systems. 2017.

---

### Official Review · AnonReviewer2 · 2019-04-15
**perceptual generative autoencoders**

**Rating:** 4
**Confidence:** 2

**Review:**

This work proposes to use autoencoders to learn perceptually meaningful spaces in which to train generative models. Two variants of the framework are introduced, using maximum likelihood training and using a variational approach. This is a fresh take on autoencoders which uses ideas from invertible neural networks to enable training of generative models in latent space.

One of the main strengths of adversarial models seems to be their ability to incorporate strong inductive biases in the loss function (i.e. the convolutional architecture of the discriminator), and this work brings that ability to other types of generative models without requiring any adversarial components (and thus neatly avoiding the instability they bring).

The manuscript would benefit from a slightly extended exposition, and possibly a few diagrams, as Section 2 was a bit hard to follow. As a reader familiar with various different autoencoder paradigms, I had a particular prior about what each component represents, and this work uses these components in new and (initially) counterintuitive ways. Emphasising these differences with prior work, and demonstrating visually where and how each component is used, would improve readability.

It would also be interesting to describe in more detail how alpha, beta, gamma are tuned, what their optimal values tend to look like, and what this means. Ablations would also be interesting. For example, what happens when beta=0? The paragraph before formula (3) provides some motivation for this particular component of the loss function, but it would be interesting to see what happens in practice.

Stating that the results in Fig. 1 are competitive with GANs is perhaps a bit of a stretch, but overall the results are promising nevertheless, and I am curious to see this idea explored further. It would especially be interesting to see if this helps with scaling up e.g. likelihood-based models and other alternatives to adversarial models.

---

### Decision · Program_Chairs · 2019-04-19
**Acceptance Decision**

Accept